# Virtual Active Learning to Maximize Knowledge Acquisition in Nursing Students: A Comparative Study

Guillermo Moreno [1,2], Alfonso Meneses-Monroy [1,*], Samir Mohamedi-Abdelkader [1], Felice Curcio [3], Raquel Domínguez-Capilla [4], Carmen Martínez-Rincón [1], Enrique Pacheco Del Cerro [1,5] and L. Iván Mayor-Silva [1]

[1] Department of Nursing, Faculty of Nursing, Physiotherapy and Podiatry, Universidad Complutense de Madrid, 28040 Madrid, Spain; guimoren@ucm.es (G.M.); samirmoh@ucm.es (S.M.-A.); nutrias@ucm.es (C.M.-R.); quique@ucm.es (E.P.D.C.); limayors@ucm.es (L.I.M.-S.)

[2] Translational Multidisciplinary Cardiovascular Research Group (ICMT), Cardiovascular Research Area, Hospital 12 de Octubre Research Institute (imas12), 28041 Madrid, Spain

[3] Department of Nursing, Faculty of Medicine and Nursing, University of Córdoba, 14004 Córdoba, Spain; felicecrc@gmail.com

[4] La Fe University and Polytechnic Hospital, 46026 Valencia, Spain; raqueldominguezcapilla94@gmail.com

[5] Processes Research Innovation and Information Systems Unit, Directorate of Nursing, Instituto de Investigación Sanitaria San Carlos (San Carlos Health Research Institute-IDISSC), Hospital Clínico San Carlos (San Carlos Clinical Hospital), 28040 Madrid, Spain

* Correspondence: ameneses@ucm.es

**Abstract:** Background: Nursing students need to acquire knowledge through active methods that promote critical thinking and decision making. The purpose of this study is to analyze whether there are differences in the acquisition of knowledge by nursing students between active face-to-face or virtual teaching methods. Methods: In this comparative study, nursing students enrolled in the psychology course were divided into two groups: a face-to-face group that received active teaching methods and a virtual group. The virtual group was exposed to the Effective Learning Strategy (ELS), which included seminars based on video content through the Virtual Campus and answering questions using the H5P tool. In addition, participants engaged in reflection tasks on the content. Covariate data were collected, and knowledge tests were administered to both groups before and after the course. After three months, subjects were re-evaluated with a final exam to assess content retention. Results: A total of 280 students were randomized. No differences were found in students' scores at the end of the knowledge test or in their final grades in the subject. Having study habits (b = 0.12, $p = 0.03$) and social support from relevant people (b = 0.09; $p = 0.03$) were associated with better post-intervention scores, and inversely with social support from friends (b = −0.12, $p < 0.01$). Final grades were inversely associated with digital safety literacy (b = −0.101, $p = 0.01$). No factors were associated with the scores of each group separately. Conclusions: The ELS virtual active learning model is as effective as face-to-face active learning methods for teaching psychology to first-year nursing students. This study was not registered.

**Keywords:** competencies; nursing; education; seminars; virtual campus; H5P

## 1. Introduction

Nursing education requires the development of critical thinking, initiative in decision making, and analytical skills [1]. Active learning methods have been shown to promote the development of critical thinking [1]. This encourages students to reflect on the content they are learning and to develop decision-making skills. Active learning generally refers to "Any instructional method that engages students in the learning process beyond listening and passive note taking. Active learning approaches focus on developing students' skills and higher-order thinking through activities such as reading, writing, and/or discussion" [2]. The learning process requires the active participation of the student as the main actor

in one's own learning [3], allowing the student to learn how to learn, which favors the development of the individual's autonomy [1,4]. In the case of nursing, active learning helps to integrate theory and practice, promotes students' self-confidence, and makes them better prepared for the job market, more empathetic, confident, and creative [1]. Active group learning develops communication skills, evaluation of individual and group learning, and awareness of individual and collective limitations and needs [5]. In addition, the use of artificial intelligence (AI) in facilitating learning outcomes has been shown to improve students' efficiency in active learning and help them solve difficult questions in test scores in combination with flipped classroom [6]. Alternative active learning methodologies, such as hands-on art and 3D atlas-based educational methods employed in anatomy education, have demonstrated notable enhancements in self-efficacy. These approaches foster creative abilities, rendering complex concepts more accessible and comprehensible [7]. Certain active learning methods have been shown to increase students' reported confidence, particularly in the acquisition of specific skills, such as bedside cardiac assessment for medical students [8]. Buzz sessions, an active learning method, make class more interesting, interactive, and help students to enhance their communication and reasoning skills and promote collaborative learning among students [9]. Clinical simulation, an active learning method commonly used in contemporary nursing education, enhances the acquisition of communication skills. Using this technique to train students in palliative care is proving effective in helping them develop meaningful relationships with end-of-life patients and their families [10]. The online problem-based active learning course, utilizing Norton's five-step process for nursing students, resulted in notable changes in educational practices. This not only contributed to enriching the academic experience but also provided opportunities for enhancing the curriculum development process, fostering collaborative learning in a community setting [11]. Recent systematic reviews indicate that active learning increases satisfaction and knowledge acquisition and generally outperforms traditional lecture-based approaches when assessed by both direct and indirect outcome measures [12]. In teaching psychology to undergraduate students using active learning methods, it has been observed that active learning is an effective tool for improving higher-level thinking and knowledge [13]. Students in perceived psychology courses responded favorably to active learning, indicating its effectiveness in achieving its intended goals. Specifically, students expressed positive views of it as an innovative approach to assessing their understanding of course content. They appreciated its role in promoting classroom interactivity, maintaining interest, engagement, and concentration, ultimately contributing to a more enjoyable learning experience [14].

The change in the COVID-19 pandemic has made it possible to implement active educational strategies based on the use of virtual platforms and the development of digital skills [15]. These platforms have been shown to encourage communication between students and teachers, and to promote self-discipline by requiring students to manage and distribute their time in order to carry out the various virtual activities proposed [16]. Despite some inconveniences associated with virtual learning, studies show that students can adapt by using protective coping strategies. This adaptation has led them to appreciate the positive elements of virtual learning, such as flexibility [17]. In a national survey of plastic surgery residents and fellows, the virtual learning format for training was found to be more time-efficient and conducive to expression of opinions compared to an in-person format [18]. Besides, some studies have shown positive effects of virtual learning on nursing knowledge, skills, and attitudes [19]. And it extends to other health care disciplines such as oral and maxillofacial surgery [20], occupational therapy, physiotherapy [21], or medicine [22]. Studies have shown that student satisfaction, performance, and evaluations are similar to those of face-to-face students when teaching psychology [23]. Moreover, online learning in teaching psychology meet the personalized requirements of the students and encourage their learning potential if it is suitable for the students' abilities [24].

The active learning methods used during the COVID-19 pandemic (H5P, infographics, videos, and scape rooms) proved to be effective in engaging nursing students in their learning process [25,26]. However, there is a lack of studies comparing the effectiveness of a face-to-face active learning methodology versus virtual active learning methodologies.

Therefore, the objectives of this study are to analyze whether there are differences in the acquisition of knowledge by nursing students between the two teaching methods.

## 2. Materials and Methods

### 2.1. Design

Comparative study of two randomized parallel groups (1:1 ratio) with students of the Bachelor of Nursing program belonging to a University in Madrid (Spain) during the academic year 2022–2023.

### 2.2. Population

All first-year undergraduate nursing students (N = 280) were included in the study. All students were enrolled in the psychology subject in which this study was conducted. Students were assigned to both groups by simple randomization in a 1:1 ratio. One group (Face-to-face Active Learning Group) received the content (which can be seen in Supplementary Table S1) under the active teaching methodology in the format of case resolution, discussion in small groups (6 participants), presentation of conclusions, explanation, and direct feedback from the teacher. The other group (Virtual Active Learning Group) received the same content virtually under a methodology that we designed and called Effective Learning Strategy (ELS). Inclusion criteria included obtaining written informed consent, while exclusion criteria included not completing all course baseline questionnaires and examinations. A total of 280 first-year nursing students were recruited. After randomization, 38 participants in the virtual group and 14 in the face-to-face group were excluded from the analysis due to noncompliance with baseline questionnaires and examinations (Figure 1).

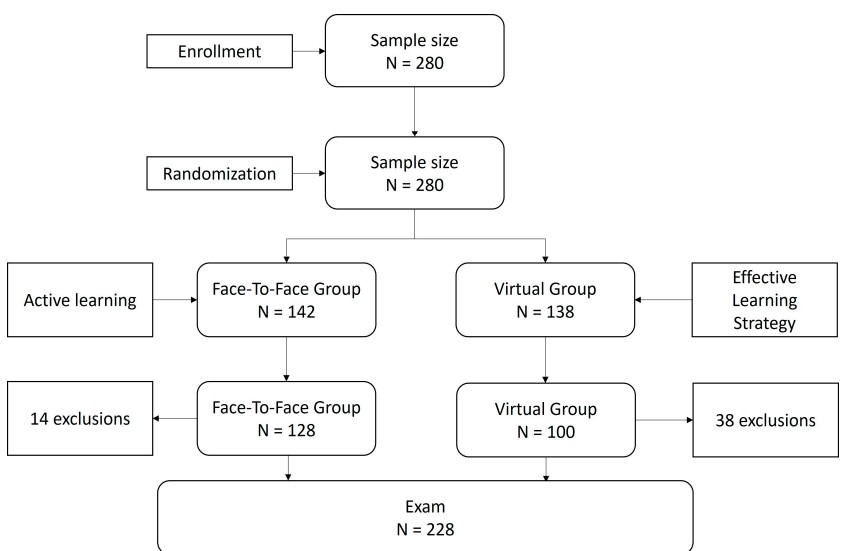

**Figure 1.** Flowchart.

### 2.3. Protocol

For the virtual group, five videos were recorded with the content of the subject and edited with the H5P tool, adding multiple choice questions (with three answer choices) related to the content in the same video. The videos were 45 min long and the questions were distributed throughout the videos every 10 min. Students were required to watch the videos and answer questions related to the content. Until the student answered the question correctly, they could not continue watching the video. At the end of watching each video, the student had to complete three tasks: summarize the main idea, justify the usefulness, and argue the application of the content to their future professional role. The aim of these tasks was to improve the student's autonomous thinking and reflection.

Both groups received the content simultaneously in different classrooms at the center. To ensure participation and viewing of the videos, the virtual active learning group was

supervised by one of the researchers. The face-to-face group was also supervised, and the students' attendance was closely monitored.

The duration of the study was five sessions of 1 h per day. A prior knowledge test was administered at the beginning of the first session, and knowledge was reassessed with the same test at the end of the last session, in each group. The test consisted of ten multiple-choice questions with three possible answers. The questions were identical for both groups (Table 1). After 3 months, the participants of both groups were evaluated on the content of the interventions in the final examination of the subject.

**Table 1.** Ten questions of the pre-post knowledge test.

(1) The stimulus that signals to a subject that his behavior will be reinforced is called:

    (a)    Excitatory conditioned stimulus.
    (b)    Discriminative stimulus (correct).
    (c)    Excitatory unconditioned stimulus.

(2) A learning process by classical conditioning in children would be… (points to True option):

    (a)    Acquire dislikes, attitudes, phobias, and fears (correct).
    (b)    Acquire study habits.
    (c)    Eliminate bad habits.

(3) Which of these varieties of learning is known as negative reinforcement?

    (a)    Escape or avoidance (correct).
    (b)    Positive punishment.
    (c)    Extinction.

(4) When the mother picks up the crying child, eliminating what for her is an annoying and unpleasant sound, this is a type of reinforcement:

    (a)    Positive reinforcement for the mother.
    (b)    Negative reinforcement for the mother (correct).
    (c)    A negative punishment for the child.

(5) Maria is a 10-year-old girl who is afraid of dogs, so she avoids going to parks where she can find them. The operant response is:

    (a)    Maria's fear when she sees a dog.
    (b)    Maria's fear of parks.
    (c)    Park avoidance behavior (correct).

(6) A learning process by classical conditioning in children would be… (points to True option):

    (a)    Acquire dislikes, attitudes, phobias, and fears (correct).
    (b)    Acquire study habits.
    (c)    Eliminate bad habits.

(7) The token economy technique is based on a program of:

    (a)    Interval.
    (b)    Reason (correct).
    (c)    Vicarious reinforcement.

(8) The subject who has a hobby of fishing (points to the True option):

    (a)    It is being reinforced by a fixed-interval program.
    (b)    It is being reinforced by a variable interval program (correct).
    (c)    It is being reinforced by a fixed-rate program.

(9) The Time-Out Technique:

    (a)    It is based on a negative reinforcement programme.
    (b)    It is based on a positive punishment programme.
    (c)    It is based on an extinction program (correct).

(10) Positive punishment consists of:

    (a)    Tell the subject about their accomplishments and positives before giving them the punishment.
    (b)    In applying a very negative stimulus when a behavior that we want to correct is emitted (correct).
    (c)    To say that punishment will have a positive purpose for the subject.

*2.4. Variables*

The independent variable was the type of active learning method (face-to-face vs. virtual). The dependent variables were the final answers to the knowledge test (Table 1) and the grades obtained in the final subject exam (we extract the qualifications from the questions of the exam related to the topics addressed in the interventions). As covariates, before the first session, sociodemographic data and psychosocial aspects were collected through different self-administered questionnaires in order to analyse their possible influence on the outcome of the intervention and on the acquisition of knowledge:

- Sociodemographic data: age, sex, and employment status.
- Learning strategies: measured by the ACRA scale. This is a self-administered instrument designed to assess learning strategies (Román and Gallego, 1994). It consists of forty-four items based on the cognitive theory and made up of three dimensions (cognitive and face-to-face learning strategies, learning support strategies and study habits). It has adequate reliability indices, $\alpha = 0.88$. It is scored on a 4-point Likert scale (1 = never, 2 = sometimes, 3 = almost always, and 4 = always). This scale has not cut-off points, the higher the student scores on each subscale, the more the student uses the learning strategy [27].
- Perceived Social Support: this variable was measured using The Multidimensional Scale of Perceived Social Support (MPSS) was administered (Zimet, 1988). It is a 12-item self-administered instrument that collects information on the individual's perception of the level of social support received in three domains: family, friends, and significant others. Each item is rated on a Likert scale (from 1—strongly disagree to 7—strongly agree). This scale has adequate psychometric indices, with an overall internal consistency of 0.89 and for each of the subscales: family (0.89), friends (0.92), and significant others (0.89). This scale does not have any cut-off points; the higher the person's score on each of the subscales, the higher the person's perceived social support [28].
- Perceived Academic stress: this variable was measured using the Academic Stressors Scale of the Academic Stress Questionnaire (ECEA) (Canabach, Valle, Rodríguez, & Piñeiro, 2008), in its latest version (Cababach, 2016), was administered. It is a self-administered instrument that assesses perceived academic stress through the degree to which major academic stressors affect college students. The scale is composed of fifty-four items grouped into eight dimensions (methodological deficiencies of the teachers, student academic overload, beliefs about academic performance, public interventions, negative social climate, examinations, content value gap, participation difficulties). Each item is rated on a 5-point Likert scale (1 = never, 2 = sometimes, 3 = quite often, 4 = almost always, and 5 = always). The scale has very good psychometric indices (total alpha of the scale of 0.96, with factors ranging from 0.79 to 0.93). This scale does not have any cut-off points; the higher the student's score on each of the subscales, the higher the student's perceived stress from that academic stressor [29].
- Perceived Digital literacy: this variable was measured using the Digital Literacy Questionnaire—IKANOS (Moscoso et al., 2022): The information collection instrument that includes the descriptors of the digComp framework validated by the European Commission in 2013. This self-administered instrument consists of 30 items related to the five competency areas analyzed by digComp (information, communication, content creation, security, and problem solving). Each item is rated on a 5-point Likert scale (1 = seldom or never, 2 = rarely or almost never, 3 = sometimes, 4 = often or almost always, 5 = very often or always). Cronbach's alpha for each of the dimensions ranged from 0.63 to 0.783. This scale has no cut-off points; the higher the person's score on each of the subscales, the higher the person's perceived digital literacy competence [30].

*2.5. Statistical Analysis*

Descriptive statistics were used by means with standard deviation or median plus interquartile range for quantitative variables and absolute and relative frequencies for qualitative variables. The Shapiro–Wilks test was used to compare normality of quantitative variables. The Student's *t*-test and the ANOVA test were used for paired and independent samples, respectively, to assess within-group differences and between-group differences in knowledge test scores and final subject exam grades. The Chi-Squared test was used for qualitative variables. Repeated measures tests (general linear model) were used to assess differences in scores obtained at each time point. Finally, the association between the covariates and the scores obtained was analysed using stepwise multiple linear regression models. Significance was defined for a 95% confidence interval and a *p*-value < 0.05. Statistical analyses were performed with the SpSS statistical tool (version 25.0).

*2.6. Ethical and Legal Considerations*

Participants' personal information was anonymized using numerical codes to ensure confidentiality. Data collection took place in October 2022, and surveys were administered anonymously. None of the researchers who participated in the data collection for the study were directly involved in the psychology course to avoid hierarchical relationships with the students. The resulting data were transcribed into a database using the anonymous identification codes previously used for each participant. The tenets of the Declaration of Helsinki on Biomedical Research Involving Human Subjects were followed at all times. Written consent to participate in the study was obtained from each student by the research team outside of class time. The voluntary and anonymous nature of the study was explained. It was also explained that their refusal to participate in the study would have no effect on the psychology subject or any other subject. This study was approved by the University Ethics Committee (internal code: CE_20220915-12_SAL).

**3. Results**

*3.1. Baseline Differences between Groups*

The face-to-face group consisted of a final sample of 128 first-year nursing students (81.2% female, 18.8% male) with a mean age of 21.08 (7.01) years; and the virtual group consisted of a final sample of 100 students (81.0% female, 19.0% male) with a mean age of 20.34 (5.43) years.

There were no differences in age (*p* = 0.39) or sex (*p* = 0.55) between the groups. Sociodemographic data are shown in Table 2.

**Table 2.** Sociodemographic differences between groups.

| | | Face-to-Face Group N (%)/M (SD) | Virtual Group N(%)/M (SD) | *p*-Value * |
|---|---|---|---|---|
| **Age** | | 21.08 (7.01) | 20.34 (5.4) | *p* = 0.39 |
| **Sex** | Woman | 104 (81.2%) | 24 (18.8%) | *p* = 0.55 |
| | Man | 24 (18.8%) | 19 (19.0%) | |
| **Employment status (working)** | | 33 (25.8%) | 19 (19.0%) | *p* = 0.15 |
| **Marital status** | Married | 95 (74.2%) | 79 (79.0%) | |
| | Single | 10 (7.8%) | 4 (4.0%) | *p* = 0.47 |
| | In a relationship (not married) | 23 (18.0%) | 17 (17.0%) | |

* Note: Chi-Squared test used.

Differences in covariate scores between the two groups were analysed to contrast whether both groups were similar in terms of learning strategies (ACRA), perceived social

support (MPSS), perceived academic stress (ECEA), and perceived digital health literacy (IKANOS). No statistically significant differences were found between the groups except for three subscales of ECEA with higher perceived stress in the face-to-face group, meaning that both groups are similar in terms of use of learning strategies, social support, and digital literacy (Table 3).

**Table 3.** Baseline differences in intergroup covariates.

| | Face-to-Face Group M (SD) | Virtual Group M (SD) | *p*-Value * |
|---|---|---|---|
| **Cognitive and Face-to-Face Learning Strategies (ACRA)** | 78.98 (10.38) | 81.02 (11.08) | *p* = 0.15 |
| **Learning Support Strategies (ACRA)** | 44.13 (6.38) | 43.79 (6.42) | *p* = 0.69 |
| **Study Habits (ACRA)** | 15.64 (3.16) | 15.60 (3.13) | *p* = 0.92 |
| **Family Social Support (MPSS)** | 23.24 (4.59) | 23.94 (4.23) | *p* = 0.24 |
| **Friends Social Support (MPSS)** | 24.16 (4.47) | 24.31 (4.58) | *p* = 0.79 |
| **Other Social Support (MPSS)** | 24.84 (4.01) | 24.40 (4.73) | *p* = 0.45 |
| **Methodological Deficiencies of Teachers (ECEA)** | 44.20 (9.98) | 41.40 (11.25) | *p* = 0.048 |
| **Student Academic Overload (ECEA)** | 34.17 (9.91) | 32.54 (10.12) | *p* = 0.22 |
| **Beliefs About Academic Performance (ECEA)** | 35.26 (11.55) | 32.24 (11.62) | *p* = 0.045 |
| **Public Interventions (ECEA)** | 26.86 (9.99) | 25.28 (9.92) | *p* = 0.24 |
| **Negative Social Climate (ECEA)** | 13.39 (6.14) | 13.75 (6.29) | *p* = 0.67 |
| **Examinations (ECEA)** | 14.16 (3.97) | 13.04 (3.99) | *p* = 0.035 |
| **Content Value Gap (ECEA)** | 10.43 (4.09) | 10.99 (4.43) | *p* = 0.32 |
| **Participation Difficulties (ECEA)** | 7.71 (3.21) | 7.41 (3.46) | *p* = 0.49 |
| **Information (IKANOS)** | 20.61 (4.77) | 20.16 (4.48) | *p* = 0.47 |
| **Communication (IKANOS)** | 20.85 (3.69) | 21.07 (3.32) | *p* = 0.64 |
| **Content Creation (IKANOS)** | 11.76 (5.20) | 12.04 (4.92) | *p* = 0.68 |
| **Security (IKANOS)** | 18.84 (4.34) | 18.35 (4.84) | *p* = 0.42 |
| **Problem Solving (IKANOS)** | 20.95 (4.14) | 20.24 (4.10) | *p* = 0.20 |

Note: ACRA: Learning strategies; MPSS: Multidimensional Scale of Perceived Social Support; ECEA: Academic Stressors Scale of the Academic Stress Questionnaire; IKANOS: Digital Literacy Questionnaire. * Student's *t*-test for independent samples.

### 3.2. Differences in Pre-Intervention, Post-Intervention, Final Scores, Within-Group, and Between-Group Scores

The differences in the pre-intervention and post-intervention scores on the knowledge tests in both groups and in the final exam scores are shown in Figure 2.

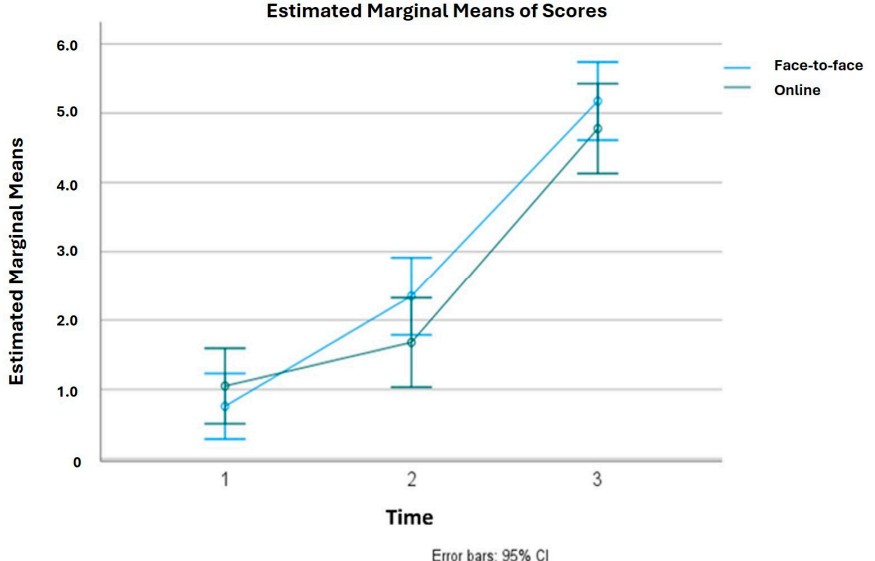

**Figure 2.** Pre-post knowledge test and final exam grade differences between groups (face-to-face and virtual). Time 1: Pre-intervention; Time 2: Post-intervention; Time 3: Final exam of the subject.

Although the face-to-face group generally showed better results, no statistically significant differences were observed between the groups in pre-intervention test scores (CG: $0.74 \pm 2.25$ vs. GI: $0.79 \pm 2.15$, $p = 0.87$), post-intervention scores (CG: $2.20 \pm 2.56$ vs. GI: $1.55 \pm 2.74$, $p = 0.10$), or final exam scores (CG: $5.12 \pm 2.46$ vs. GI: $4.75 \pm 2.76$, $p = 0.32$). Pre-post improvements were observed in both groups, with a higher mean score improvement in the face-to-face group (mean improvement in CG: 1.72; mean improvement in IG: 0.65, $p = 0.03$).

### 3.3. Repeated Measures Linear Models

In the general linear model, no statistically significant differences were observed in the interaction time by group, as the lines of both groups crossed from the pre-intervention to the post-intervention period ($p = 0.059$). There were also no significant changes between the groups from the pre-intervention to the final grades of the subject ($p = 0.198$). No overall differences were found between the two groups ($p = 0.35$) (Table 4).

**Table 4.** Repeated measures linear models.

|  |  | Sum of Squares | F | *p* |
|---|---|---|---|---|
| **Test within-subjects contrasts** | Interaction time × group (time pre vs. post) | 32.703 | 3.635 | 0.059 |
|  | Interaction time × group (time post vs. exam) | 16.776 | 1.675 | 0.198 |
| **Test of between-subjects effects** | Overall differences between groups | 380.871 | 0.878 | 0.350 |

### 3.4. Predictors of Post-Intervention and Final Grades

To evaluate the influence of the covariates on the post-intervention scores and the subjects' final grades, the variables that predicted the scores in both groups at the post-intervention and final grades were first analysed. According to the multiple linear regression model, in both groups, study habits and social support from relevant people were associated with better post-intervention scores. A non-significant trend was found for employment status (not working). This model explains 11.0% of the variability of the post-test scores (R = 0.33) with the selected variables (Table 5). Regarding the final grades, only the safety variable (IKANOS) was inversely associated with the subject's final exam scores, explaining 2.8% of the variability of the scores (R = 0.181). No associations were found in each group separately for each of the evaluation moments.

**Table 5.** Multiple linear regression model of post-intervention grades and final subject grades.

|  |  | Coefficients B | Std. Error | *p* |
|---|---|---|---|---|
| **Post-intervention** | Does not work | 0.734 | 0.383 | 0.057 |
|  | Study Habits (ACRA) | 0.116 | 0.052 | 0.026 |
|  | Social Support: Friends (MPSS) | −0.123 | 0.042 | 0.004 |
|  | Social Support: Other Relevant (MPSS) | 0.099 | 0.044 | 0.026 |
| **Exam Notes** | Security (IKANOS) | −0.105 | 0.041 | 0.012 |

Note: ACRA: Learning strategies; MPSS: Multidimensional Scale of Perceived Social Support; IKANOS: Digital Literacy Questionnaire.

## 4. Discussion

Although there is an improvement in knowledge acquisition in the experimental group between the pre-intervention and post-intervention period (mean improvement in IG: 0.65), our results of the linear model of repeated measures comparing knowledge acquisition between groups (F = 0.878; $p$ = 0.350) lead us to conclude that the virtual active learning methodology, based on conducting seminars with audiovisual material created with the H5P tool, does not allow us to obtain better knowledge acquisition. Therefore, it is not more effective as a learning strategy than the face-to-face active learning model. The results of the two methods for teaching psychology in nursing students are similar. Our results do not necessarily mean that H5P and the ELS methodology that we have developed cannot be used to teach psychology to nursing students; in fact, it can be used as it is equally effective in improving learning through knowledge acquisition. Despite the fact that the H5P tool and the virtual active learning methodology it uses have been developed as a methodology to improve active learning [31], not many studies have been conducted with students to evaluate the effectiveness of this tool/methodology. The group of Wehling et al. tested this methodology in the teaching of otolaryngology and concluded that the use of interactive H5P tools through the Moodle LMS provides a great benefit to the teaching process by allowing the easy adaptation of pre-existing video material into appropriate online content [32]. However, these authors do not provide data on the evidence of this tool in improving knowledge acquisition, and therefore our data are not comparable with this study.

According to our findings, students' knowledge would be directly related to students' study habits ($\beta$ = 0.116; $p$ = 0.026), perceived social support from relevant people ($\beta$ = 0.099; $p$ = 0.026), and inversely related to friends' support ($\beta$ = −0.123; $p$ = 0.004) and digital literacy in terms of security ($\beta$ = −0.105; $p$ = 0.012). There may also be a direct relationship between knowledge acquisition and employment status ($\beta$ = 0.734; $p$ = 0.057). Although data published in the literature indicate that students who work part-time study fewer hours, this does not affect participation in academic activities or absenteeism [33]. Our findings may differ from those of other recent studies, such as those by Warner et al., who found that employment status was not associated with poorer academic outcomes among nursing students, whereas ethnicity, race, and number of sleep hours were associated with poorer academic outcomes [34]. However, there is controversy on this point, as more classical studies have shown that nursing students who work while studying have worse academic performance, although it has been shown that academic results depend more on the number of hours worked per week than on the actual fact of working, which, as other authors have commented, would not affect part-time workers [35].

We did not find associations with the level of academic stress and the level of knowledge acquired. However, these results should be interpreted with caution, as both groups have baseline differences in perceived academic stress, with higher academic stress in some areas in the face-to-face group, which may have influenced our results.

Our findings regarding social support as a determinant of academic performance (friends' support: $\beta$ = −0.123; $p$ = 0.004 and relevant people: $\beta$ = 0.099; $p$ = 0.026) are both consistent and contradictory to other studies, which have found that low social support from friends and family predicts poor academic performance [36], whereas, according to our results, academic performance would only be associated with a lack of support from relevant people and high social support from friends. An alternative explanation for this phenomenon may be related to the use of social applications that students use to connect with their friends. The increase in the number of hours students spend on social media has a negative impact on study habits and academic performance [37,38]. Therefore, social support from friends may interfere with academic performance if students spend too much time on social networks and neglect their academic responsibilities, rather than building social support relationships, which is a predictor of good academic performance [39].

To our knowledge, there are no studies that have examined the role of digital literacy in academic performance (digital literacy in terms of security: $\beta$ = −0.105; $p$ = 0.012). However,

the use of digital learning based on gamification (which is related to high digital literacy) has been associated with better academic performance [40,41]. However, this association does not fully explain the negative association we found between digital literacy in security and academic performance, so these results should be confirmed in future studies.

### 4.1. Practical Implications

Given that there is evidence that mandatory face-to-face attendance can be counterproductive to academic performance and that student motivation is one of the most important predictors of academic success [42], the results of this study open up the possibility of working with other types of audiovisual learning methods that allow students greater freedom and, as our data show, do not result in a decrease in their academic performance. Conversely, the use of such methods is advisable in specific situations where traditional face-to-face modalities are impractical, such as cases of student overcrowding, insufficient classroom space, or a shortage of teaching staff.

### 4.2. Limitations

The data in this study came from a single academic centre and a small sample of nursing students. Other studies with more students, including other programs, and with more content should be conducted to contrast and expand our findings regarding the usefulness of using active learning methods to develop educational materials and their impact on students' academic performance. On the other hand, this study did not include student satisfaction with both methodologies, and therefore it was not possible to analyse the influence of motivation.

### 5. Conclusions

The ELS virtual active learning teaching method through video creation with H5P has shown similar effectiveness to face-to-face active learning methods in knowledge acquisition in psychology among nursing students, so its use in other disciplines could be explored and considered. This opens the possibility of implementing more virtual methods in university environments, allowing students to explore other modalities without compromising their academic performance. Future studies should investigate the influence of student motivation on knowledge acquisition in virtual and face-to-face active methods.

**Supplementary Materials:** The following supporting information can be downloaded at: https://www.mdpi.com/article/10.3390/nursrep14010011/s1, Table S1: Contents taught in both groups.

**Author Contributions:** G.M.: Conceptualization, Methodology, Writing—Reviewing and Editing. A.M.-M.: Writing—Reviewing and Editing. C.M.-R.: Data curation, Visualization, Software, Validation. R.D.-C.: Writing—Reviewing and Editing. F.C.: Writing—Reviewing and Editing. S.M.-A.: Writing—Reviewing and Editing. E.P.D.C.: Writing—Reviewing and Editing. L.I.M.-S. All authors have read and agreed to the published version of the manuscript.

**Funding:** This research received no external funding.

**Institutional Review Board Statement:** The authors declare that all procedures contributing to this work complied with the ethical standards of the relevant national and institutional committees on human experimentation and with the Helsinki Declaration of 1975 as revised in 2008. All procedures involving human subjects/patients were approved by the Ethics and Research Committee of the Faculty of Nursing, Physiotherapy and Podiatry (approval number: CE_20220915-12_SAL). All participants enrolled in the study were informed verbally and in writing about the objectives and conditions of the study. Written informed consent was obtained from all participants.

**Informed Consent Statement:** Informed consent was obtained from all subjects involved in the study.

**Data Availability Statement:** The data presented in this study are available on request from the corresponding author. The data are not publicly available due to data protection policy.

**Public Involvement Statement:** No public involvement in any aspect of this research.

**Guidelines and Standards Statement:** This manuscript was drafted against the STROBE for Observational research.

**Acknowledgments:** We would like to thank the students for their generosity and contribution.

**Conflicts of Interest:** The authors declare no conflicts of interest.

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
