# Peer review of "Virtual Active Learning to Maximize Knowledge Acquisition in Nursing Students: A Comparative Study"

_nursrep, doi:10.3390/nursrep14010011_

Round 1

Reviewer 1 Report

Comments and Suggestions for Authors

Dear all, congratulations on your idea, the work solves a pressing problem - remote work or face to face?

The work needs improvement

The type of classes conducted may affect their effectiveness. The introduction covers only general issues related to face-to-face teaching and online teaching. It seems that it should contain information about how it applies to teaching psychology. You should therefore add information or change it enough to answer this question.

 Line 121-123

the wording of the exam questions should be provided, it is not a standard tool, so it is difficult to assess what the test examined, whether it was soft or hard skills

 methodology  line 136-140

ACRA is examining learning strategies, is there a cut-off point for it? what is the interpretation of the scale?

 Line 142-146

what is the interpretation of the scale? self-efficacy? what is the result, what is the name of the indicator and what does this information provide? Is it sten interpration Me or real points?

Similarly in the subsequent scales used.

results

line 202 diagram,

 how many were examined 280 as in the diagram or 256 as in the summary? Is this a mistake? is it true. And if this is true, what happened to the remaining 24 people, and if they were excluded, why?

table 1 why do the number of women and men add up to 100.1%?

no interpretation in the text of the results of table 2, if it is an average, what does this result mean apart from the fact that we know its value, it is difficult to say anything about whether it is correct or not, no explanation in the methodology

figure 1 what does this figure refer to, are these points obtained during tests? what does she evaluate knowledge? attitudes, is this the sum of the indicators from table 2?

providing the scales from which the presented values come in the table means that the tables become difficult or impossible to interpret

line 230-234

why don't you show the whole model, only the interpertations? the model should be posted

discussion

the discussion does not refer to the results of the study, there is no explanation of specific values, nor is there any attempt at them interpretation

Is the method suitable for teaching psychology? or other items too? I didn't find such answers to my questions.

Best regards

Author Response

We thank the Editorial Board for their consideration of our manuscript. We respond below in the attached doc to each of the points raised by the reviewers.

Reviewer 2 Report

Comments and Suggestions for Authors

1. The subjects of this study appear to be nursing students taking the class. Even if approval was obtained through IRB, it seems that it is the professor conducting the research and the students taking the class. Therefore, a hierarchical relationship arises between the researcher and the subjects. It is necessary to describe the research explanation process and consent in more detail.

2. Is it possible to edit the title to make it more clear? It would be nice if the title showed the effect of the teaching method (Effective learning strategy, or virtual active leaning) applied in the comparative study.

Author Response

Editor and Reviewer comments:  

We thank the Editorial Board for their consideration of our manuscript. We respond below to each of the points raised by the reviewers.

Reviewer #2

  1. The subjects of this study appear to be nursing students taking the class. Even if approval was obtained through IRB, it seems that it is the professor conducting the research and the students taking the class. Therefore, a hierarchical relationship arises between the researcher and the subjects. It is necessary to describe the research explanation process and consent in more detail.

We thank the reviewer for his comments. Accordingly, we have changed the paragraph on ethical and legal considerations:

Participants' personal information was anonymized using numerical codes to ensure confidentiality. Data collection took place in October 2022, and surveys were administered anonymously. None of the researchers who participated in the data collection for the study were directly involved in the psychology course to avoid hierarchical relationships with the students. The resulting data were transcribed into a database using the anonymous identification codes previously used for each participant. The tenets of the Declaration of Helsinki on Biomedical Research Involving Human Subjects were followed at all times. Written consent to participate in the study was obtained from each student by the research team outside of class time. The voluntary and anonymous nature of the study was explained. It was also explained that their refusal to participate in the study would have no effect on the psychology subject or any other subject. This study was approved by the University Ethics Committee (internal code: CE_20220915-12_SAL).

  1. Is it possible to edit the title to make it more clear? It would be nice if the title showed the effect of the teaching method (Effective learning strategy, or virtual active leaning) applied in the comparative study.

We thank the reviewer for his suggestion. We have changed the title to include his suggestion:

Virtual active learning to maximize knowledge acquisition in nursing students: a comparative study

Reviewer 3 Report

Comments and Suggestions for Authors

Firstly, I'd like to express my gratitude for allowing me to review this article. I found it very interesting and in line with the current trends in higher education, where alternative forms of learning beyond in-person instruction are being explored.

As reflected in the limitations of this study, the satisfaction of students with different teaching methods has not been addressed. It would have been very interesting to analyze this variable as well.

Author Response

Editor and Reviewer comments:  

We thank the Editorial Board for their consideration of our manuscript. We respond below to each of the points raised by the reviewers.

Reviewer #3

Firstly, I'd like to express my gratitude for allowing me to review this article. I found it very interesting and in line with the current trends in higher education, where alternative forms of learning beyond in-person instruction are being explored.

We want to thank the reviewer for his/her interest and for appreciating our work.

As reflected in the limitations of this study, the satisfaction of students with different teaching methods has not been addressed. It would have been very interesting to analyze this variable as well.

We thank the reviewer for his suggestion. However, as the reviewer points out in his comment, we did not include this variable in the design of our manuscript and unfortunately we cannot provide any results in this sense.

Reviewer 4 Report

Comments and Suggestions for Authors

An interesting topic, particularly after the Covid-19 pandemic, as it shows that the use of virtual learning models should be explored and can help to overcome difficulties that currently arise in teaching and learning processes.

1. Introduction: Adequate. Explores concepts and draws on current literature related to the topic. Justifies the study. Aim well.

2. Materials and Methods: Well described, with the exception of the following aspects that can be improved:

2.2. Population: Figure 1. Flowchart. It should be in this subchapter. It would help to explain how to obtain the number of participants; however, the group randomization process is not presented, which must be done.

2.5. Statistical analysis: does not present an adopted level of significance.

3. Results: the reference to the results in "Table 2. Baseline differences in intergroup covariates." is not correct. In ECEA there are statistically significant differences in 3 dimensions.

The statistical tests used are not identified.

4. Discussion: Adequate and sufficiently supported. The "Practical implications" are highlighted.

5. Conclusions: Very brief, would benefit from being further explained.

References: Current and appropriate.

Author Response

Editor and Reviewer comments:  

We thank the Editorial Board for their consideration of our manuscript. We respond below to each of the points raised by the reviewers.

Reviewer #4

An interesting topic, particularly after the Covid-19 pandemic, as it shows that the use of virtual learning models should be explored and can help to overcome difficulties that currently arise in teaching and learning processes.

  1. Introduction: Adequate. Explores concepts and draws on current literature related to the topic. Justifies the study. Aim well.

We want to thank the reviewer for his/her interest and for appreciating our work.

  1. Materials and Methods: Well described, with the exception of the following aspects that can be improved:

2.2. Population: Figure 1. Flowchart. It should be in this subchapter. It would help to explain how to obtain the number of participants; however, the group randomization process is not presented, which must be done.

We thank the reviewer for his suggestion. Accordingly, we have moved Figure 1 and the paragraph describing allocation and exclusion to this paragraph.

2.5. Statistical analysis: does not present an adopted level of significance.

We thank the reviewer for this comment. Accordingly, we have added this information to the statistical analysis section:

Significance was defined for a 95% confidence interval and a p-value <.05.

  1. Results: the reference to the results in "Table 2. Baseline differences in intergroup covariates." is not correct. In ECEA there are statistically significant differences in 3 dimensions.

We thank the reviewer for his/her suggestion. Accordingly, we have added the following paragraph to the Results section to explain the objectives of this contrast analysis and their significance:

Differences in covariate scores between the two groups were analyzed to contrast whether both groups were similar in terms of learning strategies (ACRA), perceived social support (MPSS), perceived academic stress (ECEA), and perceived digital health literacy (IKANOS). No statistically significant differences were found between the groups except for three subscales of ECEA with higher perceived stress in the face-to-face group, meaning that both groups are similar in terms of use of learning strategies, social support, and digital literacy (Table 3).

We have also added the following paragraph to the discussion section:

We did not find associations with the level of academic stress and the level of knowledge acquired. However, these results should be interpreted with caution, as both groups have baseline differences in perceived academic stress, with higher academic stress in some areas in the face-to-face group, which may have influenced our results.

The statistical tests used are not identified.

We thank the reviewer for his/her feedback. We have included the statistical test used in the tables where we present our results. Other tests are described in the Methodology section:

Table 2. Sociodemographic differences between groups.

Face-to-face group

N (%)/M (SD)

Virtual Group

N(%)/M(SD)

p-value*

Age

21,08 (7,01)

20,34 (5,4)

P = 0.39

Sex

Woman

104 (81,2%)

24 (18,8%)

P = 0.55

Man

24 (18,8%)

19 (19,0%)

Works

33 (25,8%)

19 (19,0%)

P = 0.15

Marital status

Married

95 (74,2%)

79 (79,0%)

P = 0,47

Bachelor

10 (7,8%)

4 (4,0%)

Common-law marriage

23 (18,0%)

17 (17,0%)

*Note: Chi-Squared test used

Table 3. Baseline differences in intergroup covariates.

Face-to-face group

M (SD)

Virtual Group

M (SD)

p-value*

Cognitive and Face-to-Face Learning Strategies (ACRA)

78,98 (10,38)

81,02 (11,08)

P = 0,15

Learning Support Strategies (ACRA)

44,13 (6,38)

43,79 (6,42)

P = 0,69

Study Habits (ACRA)

15,64 (3,16)

15,60 (3,13)

P = 0,92

Family Social Support (MPSS)

23,24 (4,59)

23,94 (4,23)

P = 0,24

Friends Social Support (MPSS)

24,16 (4,47)

24,31 (4,58)

P = 0,79

Other Social Support (MPSS)

24,84 (4,01)

24,40 (4,73)

P = 0,45

Methodological Deficiencies of Teachers (ECEA)

44,20 (9,98)

41,40 (11,25)

P = 0,048

Student Academic Overload (ECEA)

34,17 (9,91)

32,54 (10,12)

P = 0,22

Beliefs About Academic Performance (ECEA)

35,26 (11,55)

32,24 (11,62)

P = 0,045

Public Interventions (ECEA)

26,86 (9,99)

25,28 (9,92)

P = 0, 24

Negative Social Climate (ECEA)

13,39 (6,14)

13,75 (6,29)

P = 0,67

Examinations (ECEA)

14,16 (3,97)

13,04 (3,99)

P = 0,035

Content Value Gap (ECEA)

10,43 (4,09)

10,99 (4,43)

P = 0,32

Participation Difficulties (ECEA)

7,71 (3,21)

7,41 (3,46)

P = 0,49

Information (IKANOS)

20,61 (4,77)

20,16 (4,48)

P = 0,47

Communication (IKANOS)

20,85 (3,69)

21,07 (3,32)

P = 0,64

Content Creation (IKANOS)

11,76 (5,20)

12,04 (4,92)

P = 0, 68

Security (IKANOS)

18,84 (4,34)

18,35 (4,84)

P = 0,42

Problem Solving (IKANOS)

20,95 (4,14)

20,24 (4,10)

P = 0,20

Note: ACRA: Learning strategies; MPSS: Multidimensional Scale of Perceived Social Support; ECEA: Academic Stressors Scale of the Academic Stress Questionnaire; IKANOS: Digital Literacy Questionnaire. * Student's t-test for independent samples

  1. Discussion: Adequate and sufficiently supported. The "Practical implications" are highlighted.

We want to thank the reviewer for his/her interest and for appreciating our work.

  1. Conclusions: Very brief, would benefit from being further explained.

We thank the reviewer for his/her feedback. We have expanded our conclusions sections with the main implications of our studies and the proposed future lines:

The ELS virtual active learning teaching method through video creation with H5P has shown similar effectiveness to face-to-face active learning methods in knowledge acquisition in psychology among nursing students, so its use in other disciplines could be explored and considered. This opens the possibility of implementing more virtual methods in university environments, allowing students to explore other modalities without compromising their academic performance. Future studies should investigate the influence of student motivation on knowledge acquisition in virtual and face-to-face active methods.

References: Current and appropriate.

We want to thank the reviewer for his/her interest and for appreciating our work.

Round 2

Reviewer 1 Report

Comments and Suggestions for Authors

After improvement, the work is suitable for printing

Reviewer 2 Report

Comments and Suggestions for Authors

Please edit table1 to make it more readable.